# DENSE CORRELATION FIELDS FOR MOTION MODELING IN ACTION RECOGNITION

## ABSTRACT

The challenge of action recognition is to capture reasoning motion information. Compared to spatial convolution for appearance, the temporal component provides an additional (and important) clue for motion modeling, as a number of actions can be reliably recognized based on the motion information. In this paper, we present an effective and interpretable module, Dense Correlation Fields (DCF), which builds up dense visual correlation volumes at the feature level to model different motion patterns explicitly. To achieve this goal, we rely on a spatially hierarchical architecture that preserves both fine local information provided in the lower layer and the high-level semantic information from the deeper layer. Our method fuses spatial hierarchical correlation and temporal long-term correlation, which is better suited for small objects and large displacements. This module is extensible and can be plugged into many backbone architectures to accurately predict object interactions in the video. DCF shows consistent improvements over 2D CNNs and 3D CNNs baseline networks with 3.7% and 3.0% gains respectively on the standard video action benchmark of SSV1.

## 1 INTRODUCTION

Action recognition is a fundamental problem in video understanding (Karpathy et al., 2014; Laptev et al., 2008). Unlike image classification, action recognition should distinguish visual tempo variation as well as its semantic appearance. Recently, great progress has been made by deep learning based models to improve the accuracy of video action recognition (Feichtenhofer et al., 2019; Jiang et al., 2019; Yang et al., 2020a). CNNs for video understanding has been extended with the capability of capturing not only appearance information contained in individual frames but also motion information extracted from the temporal dimension of the image sequence.

One common method for action recognition is to use a two-stream network (Simonyan & Zisserman, 2014; Crasto et al., 2019; Feichtenhofer et al., 2016; Qiu et al., 2019), where one stream is on raw frames to extract appearance information, and the other is to leverage optical flow to learn motion information. An alternative strategy implicitly uses 3D CNNs (Carreira & Zisserman, 2017; Tran et al., 2015a; Feichtenhofer et al., 2019) or temporal convolution (Tran et al., 2018; Xie et al., 2018), as these methods can jointly capture spatial and temporal information in a unified spatiotemporal framework. Some other methods extend 2D CNN-based backbones with temporal modules (Lin et al., 2019; Li et al., 2020b; Meng et al., 2021; Liu et al., 2021b) to learn motion information. However, the performance of previous action recognition systems is limited by difficulties including small objects and large displacements (fast moving objects). One conundrum for these methods is that the high-level feature is semantically strong but spatially coarse, as the spatial feature is crucial to capture motion information. As the cases shown in Figure 1, strong semantic features and fine spatial features are both the keys to distinguishing action classes.

This paper propose Dense Correlation Fields (DCF), a new temporal module for motion modeling. The correlation operator captures the motion information by computing the alignment of visually similar image regions between frames. Visual correlation is highly relevant to optical flow, which is the most important clue for capturing motion patterns. The correlation operator can be used as approximate motion information, which has shown effectiveness in action recognition in CorrNet (Wang et al., 2020). DCF combines low-resolution, semantically strong correlation features with high-resolution, semantically weak correlation features to recover different motion patterns. Our

(a) Small object (the pen on the box)

(b) Moving something at a fast tempo

Figure 1: The action examples above show small object and large displacement (fast moving object) from SSV1 valid videos. (a) This example is captioned with 'Moving something across a surface until it falls down'. There is another similar type of action where the object does not fall down. The action of the pen on the box is crucial to distinguish these two classes. (b) This example shows a fast-tempo action with large displacement across the frames.

DCF consists of two main contributions: (1) correlation aggregation over a spatial pyramidal hierarchy; (2) cross-frame correlation volume with both short-term and long-term temporal information. Our DCF enables the efficient integration of spatial information and semantic information for motion modeling throughout the network.

The design of DCF draws inspiration from many existing works but is substantially novel. First, DCF builds up dense fields by combining features from spatial pyramidal correlation hierarchy. This is different from the individual temporal module applied over multi-stage in prior works (Wang et al., 2020; 2018b; Huang et al., 2021). Temporal modeling normally presents short-term motion between adjacent frames at low-level feature and long-term temporal aggregation at high-level feature. These methods use motion modeling as different stages to deal with different motions. In practice, this fails in cases where the determined motion on a lower scale is too spatially coarse to be close to the correct motion of a higher scale. Our DCF uses spatial pyramid hierarchy to hallucinate spatially coarser but semantically stronger correlation feature by spatially finer correlation feature. The principle advantage of featuring each level of a correlation pyramid is that it produces a multi-scale motion feature representation in which all levels are spatially fine, including the low-resolution levels.

Second, DCF maintains a cross-frame correlation with both short-term and long-term temporal information. While CorrNet only uses the correlation between adjacent frames, we compute the correlation between consecutive frames to form long-term temporal information, following the strategy of previous methods (Wang et al., 2021; 2018b). DCF provides the network with long-term motion information by operating on cross-frame correlation volume.

DCF can be applied to different backbone architectures as a plugin module. We construct DCF networks with two backbone networks, (R(2+1)D (Tran et al., 2018) and X3D(Feichtenhofer, 2020)). In order to evaluate the proposed method in terms of modeling motion variations, we construct experiments on the Something-Something dataset (Goyal et al., 2017) which has been well-known to be challenging to classify an action due to the temporal complexity. In addition, we validate various design choices of DCF through extensive ablation studies. Moreover, we show the performance on Kinetics-400 dataset (Kay et al., 2017) to compare the proposed method to the many state-of-the-arts.

## 2 RELATED WORK

**Action Recognition.** Action recognition research has been largely driven by learned features and various learning models utilizing deep networks. Two-stream CNNs (Simonyan & Zisserman, 2014) with one stream of static images and the other stream of optical flows are proposed to fuse the information of appearance and motion. Temporal Segment Networks (Wang et al., 2016) sample frames and optical flow on different time segments to extract information for activity recognition. 3D-CNNs (Ji et al., 2012; Tran et al., 2015b) proposed 3D convolution to directly learn spatiotemporal features from videos. Several variants decompose 3D convolution into a 2D convolution and a 1D temporal convolution, for example P3D Qiu et al. (2017), R(2+1)D (Tran et al., 2018), S3D (Xie et al., 2018), and CT-Net (Li et al., 2021). Recently, the great success of image Transformers has led to investi-

gation of Transformer-based architectures for video recognition tasks (Liu et al., 2021a; Bertasius et al., 2021; Arnab et al., 2021; Fan et al., 2021; Neimark et al., 2021) Our work is motivated by the success in incorporating optical flow for action recognition. There is an important issue in existing methods with optical flow for action recognition: the dependency on beforehand extraction of optical flow lowers the efficiency and effectiveness of the recognition system. Our DCF module inherits the effectiveness of optical flow and provides the video model with precise motion information.

**Motion Modeling for Action Recognition.** For action recognition, the appearance of still frames and motion information such as optical flow is the most important cues to identify the action. Many recent works (Choutas et al., 2018; Sun et al., 2018; Jiang et al., 2018; Fan et al., 2018; Asghari-Esfeden et al., 2020; Wu et al., 2018) design powerful temporal modules and insert them into 2D CNNs for efficient action recognition. TRN (Zhou et al., 2018) and TSM (Lin et al., 2019) capture information along the temporal dimension with an interpretable relational module and a shift module respectively. Some methods learn motion information by incorporating optical flow or RGB difference for action recognition systems. TDN (Wang et al., 2021) generalizes temporal difference operator for capturing both short-term and long-term temporal. ActionFlowNet (Ng et al., 2018) proposes to jointly estimate optical flow and recognize actions in one network. Several works propose carefully designed modules to capture different types of temporal dependency, such as Slowfast (Feichtenhofer et al., 2019), TPN (Yang et al., 2020a), Non-local Net (Wang et al., 2018b).

Some methods(Zhao et al., 2018; Wang et al., 2018a) introduce cost volume or multiplicative interactions without the reliance on optical flow. CorrNet(Wang et al., 2020) and MSNet(Kwon et al., 2020) also compute frame-to-frame correlation over feature maps for effective motion estimation. (Piergiovanni & Ryoo, 2019) propose a flow layer that unrolls the iterations of the TV-L1 algorithm with learned parameters. DynamoNet(Diba et al., 2019) propose dynamic motion filters by predicting the future frames to enrich motion representation. However, these methods learn motion representation at single level, or multiple levels of the network individually, which may encounter a problem that local information disappears at spatially coarse levels. (Yang et al., 2020b) also present hierarchical method to bridge different levels in a network. This hierarchical design use discriminative contrastive loss to enforce the motion features at high-level to predict the ones at low-level. However, this contrastive learning encourages features to have more similar representations, leading to less diversity to cover motion patterns. In contrast, we combine low-level features with high-level features via top-down and skip connections. In this way, we can preserve the diversity from the fine local information provided in the lower layer and the high-level semantic information from the deeper layer.

## 3 APPROACH

The Dense Correlation Fields are designed to model motion patterns explicitly. The proposed DCF can be applied to different backbone architectures as a plugin module. An overview of our approach is given in Figure 2. Given a video clip, we sample $T$ frames as input I with the shape of I is $[T, H, W, D]$, where $D$ is the number of channels and $H, W$ are the spatial resolution. The backbone network is applied to I and maps the input frames to feature maps at lower spatial resolution. This process is independent of the backbone architectures, and here we present a backbone example using ResNets(He et al., 2016). Typically, a ResNet-like backbone outputs feature pyramid at 4 resolutions for multiple stages, and note that they have spatial strides of $4, 8, 16, 32$. We construct Dense Correlation Fields by exploiting connections that associate correlation volume across multiple stages. DCF is built upon a collection of spatial pyramid correlation features. We first detail the specific instantiation of the correlation operator, which computes frame-to-frame visual similarity. Afterwards, we show how our DCF aggregates motion information over the correlation features from a spatial pyramid feature hierarchy. Finally, provide the implementation detail to instantiate DCF with backbone networks.

### 3.1 BASIC CORRELATION OPERATOR

The correlation operator explicitly captures the motion information by computing visual similarity between two frames. Given two frame features $f_t \in \mathbb{R}^{H \times W \times D}$ and $f_{t+\tau} \in \mathbb{R}^{H \times W \times D}$, the correlation operator volume is formed by taking the dot product between feature vectors. For a feature vector $f_t(i, j)$, we compute the similarity of this vector with vectors in $f_{t+\tau}$, where $(i, j)$ is the

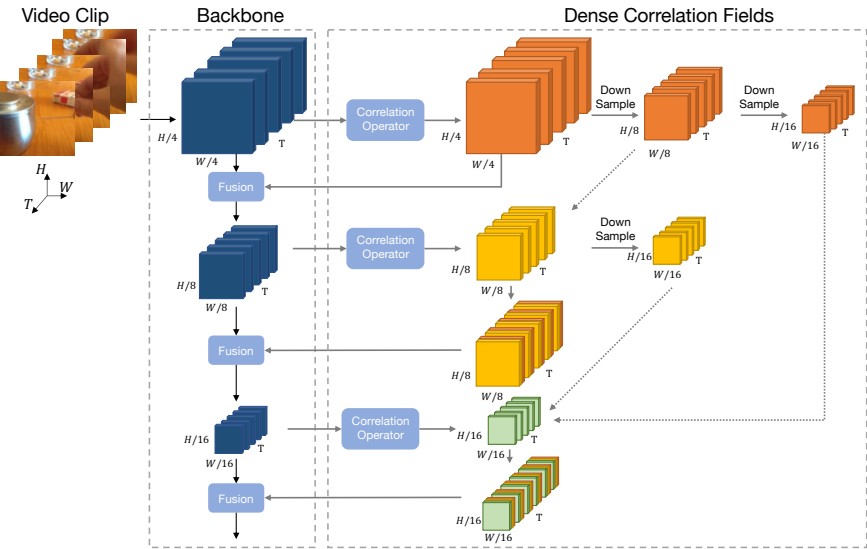

Figure 2: The framework of Dense Correlation Fields with spatial pyramid feature hierarchy. A Backbone network extracts multi-level features for a video clip. A correlation layer constructs a 4D correlation volume with shape $H \times W \times T \times (K \times K \times 2L)$ for each level feature. K indicates the spatial region and L indicates the temporal range. Correlation volumes in the low-level stages are down-sampled to generate multi-scale volumes, which are connected to high-level stages. The volumes from the current stage and the bottom-up pathway are combined to learn motion information. A learned block aggregates the motion information from the correlation hierarchy. Our dense correlation fields maintain coarse and fine resolution information, also long-term and short-term temporal information. It helps to handle various object sizes and motion tempos in action recognition.

spatial location of the vector. We consider pixel-to-pixel similarity between two frames, $f_t$ and $f_{t+\tau}$,

$$\mathrm{C}(f_t, f_{t+\tau}) \in \mathbb{R}^{H \times W \times K \times K}, \mathrm{C}_{i,j} = \frac{1}{\sqrt{D}} \left\langle f_t(i,j), f_{t+\tau}(i',j') \right\rangle \tag{1}$$

where $\frac{1}{\sqrt{D}}$ is for normalization. $(i',j')$ is often limited in a $K \times K$ spacial neighborhood of $(i,j)$ for computational reason. A correlation volume can be seen as a set of visual similarities between the pairs of consecutive frames $t$ and $t + \tau$.

## 3.2 DENSE CORRELATION FIELDS

In this subsection, we describe Dense Correlation Fields, which improve the action recognition system's robustness to capture motion variation. We first consider an extension of correlation from image-wise to video-wise. This aims to better capture long-term dependencies. Then we build a spatial pyramid correlation features on top of it. Our goal is to preserve both fine spatial information provided in the lower layer and the high-level semantic information from the deeper layer.

**Long-term Correlation.** To represent the motion across a sequence of $T$ frames, we first repeat the correlation operator in video frames by computing the correlation volume for every pair of adjacent frames of the input sequence, obtain video-level correlation volume $\mathrm{S} = [S_1, ..., S_T]$. It is natural to consider an extension to a bi-directional correlation, which can be obtained by computing an additional set of displacements in the opposite direction. We stack the forward correlation between frames $t$ and $t + 1$ and the backward correlation between frames $t$ and $t - 1$. To capture long-range dependencies, we construct long-term correlation across $2L$ neighborhood frames ($L$ forward frames and $L$ backward frames). We enlarge the matching region as the temporal step increases by a dilation factor. In this long-range bi-directional case, we combine multiple displacement fields into a single correlation volume by concatnation,

$$\mathrm{S}_t = \{\mathrm{C}(f_t, f_{t-L}), ..., \mathrm{C}(f_t, f_{t-1}), \mathrm{C}(f_t, f_{t+1}), ..., \mathrm{C}(f_t, f_{t+L})\} \tag{2}$$

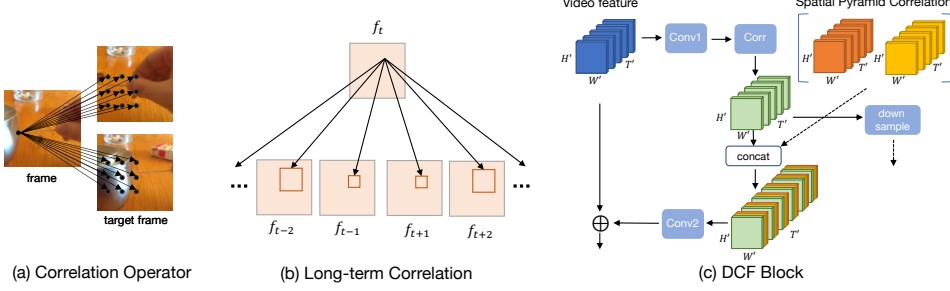

Figure 3: Dense Correlation Fields detail. (a). The correlation is formed by taking the inner product of feature vectors between patches from different frames. (b). In order to explore the long-term temporal dependency, we construct correlation volume between bidirectional and consecutive frames. (c). The feature of the current stage first undergoes a $1\times1\times1$ layer to reduce channel dimensions and then compute a correlation volume. The correlation volume of current stage is combined with correlation volumes from the top-down pathway. These correlation volumes are attached by another $1\times1\times1$ Conv (we omit BN and ReLU for clarity) to produce the correlation feature. The video feature is merged by addition with the correlation feature. The current correlation volume is downsampled spatially and connected to the deeper stage for a pyramid correlation.

Thus, the correlation volume S for $T$ frames has dimensions $T \times H \times W \times (K \times K \times 2L)$ (last dimension can be seen as correlation volume channels).

**Spatial Correlation Hierarchy Aggregation.** We employ the correlation block for multiple stages and obtain the correlation volumes as $S^i$ at several spatial scales with a scaling step of $2^i$. We further construct a spatial correlation pyramid by collecting correlation volume hierarchy. For example, we have $P^4$ for the correlation pyramid in the stage 4,

$$P^4 = \{S^4, \text{Pool}(S^3), \text{Pool}^2(S^2)\} \tag{3}$$

where $\text{Pool}^k()$ denote pooling at the spatial dimensions with kernal $2^k$. The pooled correlation volume $\text{Pool}^k(S)$ has dimensions $T \times H/2^k \times W/2^k \times (K \times K \times 2L)$.

We use a top-down pathway to combine correlation volume from lower resolution with spatially stronger correlation volume from higher resolution. We wrap our DCF into a residual connection block that can be incorporated into many existing architectures. The residual connection allows us to insert our DCF block into any pre-trained model, without breaking its initial behavior. Figure 3.(c) shows the building block that constructs our DCF. The video feature first undergoes a $1\times1\times1$ layer to reduce channel dimensions and then is attached a correlation layer. The correlation volume is then merged with the top-down correlation volumes by concatenation in the channel dimension. Finally, we append another $1\times1\times1$ layer (we omit BN and ReLU for clarity) on the stacked correlation volumes to generate the final feature map, which is merged with the input video feature by element-wise addition. To generate a spatial correlation hierarchy, we simply downsample (using average pooling) the spatial resolution of the current correlation volume for deeper stage. The top-down pathway starts at early stage without correlation volume input. This process is iterated until the final dense correlation fields are generated.

## 3.3 DCF NETWORKS

Our DCF presents a new motion modeling mechanism to learn action explicitly in action recognition tasks. DCF is designed as a plug-in substitution for video models. We describe our baseline network architecture for this task, and then wrap them into our proposed DCF networks. In practice, the DCF only adds a small overhead to the computational cost of the backbone network.

We first use the R(2+1)D (Tran et al., 2018) with ResNet (He et al., 2016) pretrained on ImageNet (Deng et al., 2009) as our backbone model to instantiate our DCF networks. Following the practice in previous temporal module methods (Wang et al., 2021; Yang et al., 2020a; Wang et al., 2020), we use the output features of $stage2$, $stage3$, $stage4$ to build DCF, where they are spatially downsampled by respectively 4, 8 and 16 times, compared to the input frames. We omit $stage5$ as its spatial resolution is too low. Compared with the original R(2+1)D, we make a few changes to further

simplify and improve its efficiency, *e.g.*, using depthwise temporal 1D-Conv, removing temporal downsampling. Alternatively, we also construct DCF with 3D CNNs based on the recent X3D model(Feichtenhofer, 2020). For the X3D backbone, we build DCF on the collection of feature maps from the stages with the same resolutions as in R(2+1)D model. For the structure detail of the R(2+1)D ResNet and X3D, please refer to Appendix.

The computation of a DCF block is lightweight when it is used in sub-sampled feature maps rather than high-resolution image. We set the channels of the first $1\times1\times1$ layer to be half of the input channels in X3D and a quarter in R(2+1)D. Then, we perform the correlation operation with spatial region size $K = 5$ and temporal range $L = 2$ ($L = 1$ for the last stage). After that, another $1\times1\times1$ layer is used to restore the original channels. We add BatchNorm (BN) (Ioffe & Szegedy, 2015) and ReLU (Nair & Hinton, 2010) layer after the second $1\times1\times1$ layers.

## 4    EXPERIMENTS ON ACTION RECOGNITION

In this section, we present the experiment results of our DCF on three large-scale public datasets. To understand the behavior of DCF, we first perform comprehensive studies on the challenging temporal-related dataset Something-Something V1 (Goyal et al., 2017). Ablation studies present consistent improvements, which show the effectiveness and generality of our DCF. We also report results on the Something-Something V2 (Goyal et al., 2017) and Kinetics dataset (Kay et al., 2017) to show the generality of our method.

### 4.1    DATASET AND IMPLEMENTATION DETAILS

**Datasets.** Something-Something (Goyal et al., 2017) is a large-scale dataset created by crowdsourcing. The videos are collected by performing the same action with different objects so that action recognition is expected to focus on the motion property instead of appearance. The first version consists of 86k training videos and 11k validation videos belonging to 174 action categories, whose durations vary from 2 to 6 seconds. The second release (V2) of Something-Something increases the number of videos to 220k. Kinetics-400 (Kay et al., 2017) is among the most popular datasets for video classification. It contains around 240k training YouTube videos and 19k validation videos that last for 10 seconds. It includes 400 action categories in total. The Kinetics dataset contains activities in daily life and some categories are highly correlated with interacting objects or scene context.

**Training and evaluation.** In R(2+1)D backbone experiments, we use ResNet50 and ResNet101 initialized using ImageNet (Deng et al., 2009) pre-trained weights (He et al., 2016) to implement our DCF networks. We train the R(2+1)D model for 60 epochs on Something-Something dataset and 100 epochs for Kinetis-400 dataset. As for X3D backbone experiments, we use X3D-S and X3D-M to train our DCF networks from scratch. We train the X3D model for 128 epochs on Something-Something dataset and 256 epochs for Kinetis-400 dataset. Following common practice (Feichtenhofer et al., 2019; Wang et al., 2021), during training, each video frame is resized to have shorter side in $[256, 320]$ and a crop of $224 \times 224$ is randomly cropped. Temporally, we perform interval based sampling for Kinetics-400, with interval of 8 for 8 frames, interval of 6 for 13 frames and interval of 5 for 16 frames. On Something-Something dataset, we perform segment based sampling with segments of 8 or 16. We implement two kinds of testing scheme. When we consider the efficiency, 1-clip per video and the center crop of $224 \times 224$ is used for evaluation on SSV1 and SSV2 dataset. When pursuing high accuracy, we followed the common setting in (Wang et al., 2018b) to sample multiple clips per video and use three spatial crops of $256 \times 256$ on Kinetics-400 dataset. We include further training details in the appendix.

### 4.2    ABALATION STUDY

This section provides comprehensive ablation studies to verify the effectiveness of the proposed DCF in terms of motion modeling. Something Something V1 dataset is used as it is widely acknowledged to focus on motion modeling. For these evaluations, we use the testing scheme of 1 clip and center crop, and report the Top1 accuracy. We first explore the effect of different components and the locations for DCF, using X3D-S as backbone network. Then we investigate the generality of DCF on different backbones.

| Basic Corr | Long Term | Spatial Pyramid | Top1 | Δ |
|:---:|:---:|:---:|:---:|:---:|
| | | | 44.6 | |
| ✓ | | | 46.1 | +1.5 |
| ✓ | ✓ | | 46.6 | +2.0 |
| ✓ | | ✓ | 47.3 | +2.7 |
| ✓ | ✓ | ✓ | **47.6** | **+3.0** |

Table 1: Contributions of the proposed components in DCF, including basic correlation operator, long-term temporal correlation and spatial pyramid aggregation.

| Stage | | | | FLOPs | Top1 |
|:---:|:---:|:---:|:---:|:---:|:---:|
| 1 | 2 | 3 | 4 | | |
| | | | | 2.0G | 44.6 |
| ✓ | | ✓ | | 2.4G | 46.5 |
| | ✓ | | ✓ | 2.1G | 46.8 |
| ✓ | ✓ | ✓ | | 2.5G | 47.2 |
| | ✓ | ✓ | ✓ | 2.2G | **47.6** |

Table 2: Different stage groups employ DCF. The results imply that employing blocks at stages 2-4 obtains the best recognition accuracy and the computational cost is also reasonable.

| Backbone | DCF | Frames | Params | FLOPs | Top1 | Δ |
|:---:|:---:|:---:|:---:|:---:|:---:|:---:|
| X3D-S | ✗ | 13 | 3.3M | 2.0G | 44.6 | - |
| | ✓ | 13 | 3.4M | 2.2G | 47.6 | +3.0 |
| X3D-M | ✗ | 16 | 3.3M | 4.7G | 47.3 | - |
| | ✓ | 16 | 3.4M | 5.2G | 49.5 | +2.2 |
| R(2+1D) R50 | ✗ | 8 | 23.9M | 32.7G | 46.8 | - |
| | ✓ | 8 | 24.6M | 35.9G | 50.5 | +3.7 |

Table 3: Different backbones employing DCF.

**Different components of DCF.** We study the effect of the individual component of DCF and the results are shown in Table 1. In particular, we measure the effect of: using basic correlation operator; long-term temporal correlation; spatial pyramid aggregation. We add one correlation block to stages of $stage2$, $stage3$, $stage4$ respectively. First of all, we can conclude that the correlation operator is beneficial for action recognition, as it provides the network with explicit motion information. Second, we find that long-term temporal correlation is also helpful, as it enhances the original representation with long-term motion information. Finally, we note that spatial pyramid correlation significantly boosts the performance by ∼1%, which confirms the importance of associating low-level correlation across resolutions and semantic levels for motion modeling. It hallucinates lower resolution features by downsampling spatially finer, but semantically weaker, correlation feature maps from higher resolution features. In the rest of this paper, we use the full version of DCF by default.

**Different stages employing DCF.** We perform the ablation study on which stage to use the correlation block. The results are shown in Table 2. First, we see that adding more correlation blocks into the backbone network will lead to better results. More blocks increase the computational cost slightly. The setting of using blocks in stages 2-4 obtains the best recognition accuracy and the computational cost is also reasonable. Noting that DCF on stage3 contributes more to the performance, as this stage provides feature maps of high-level semantics as well as more accurately localized information. In the rest of this paper, we use three blocks (1 to $stage2$, 1 to $stage3$, 1 to $stage4$) to build DCF by default.

**DCF with different backbones.** We compare the proposed DCF with other backbone architectures in Table 3. DCF improves the recognition performance with small computation overhead on different backbones, including X3D (Feichtenhofer, 2020) and R(2+1)D (Tran et al., 2018) by 3.0% (2.2% for X3D-M) and 3.7% respectively on SSV1. It is noteworthy that the improvement of DCF is not just because they add depth to the baseline model. To see this, we note that in Table 3 the DCF with X3D-S model has 47.6% accuracy, higher than the deeper X3D-M baseline's 47.3%. This comparison shows that the improvement due to DCF is complementary to going deeper in standard ways.

### 4.3 COMPARISON WITH THE STATE OF THE ART

After the ablation study of DCF on Something-Somthing V1 dataset, we directly transfer its optimal setting to the datasets of Something-Something V2 and Kinetics-400. Compared to Kinetics-400, Something-Something V1&V2 require more temporal modeling ability than spatial appearance. In this subsection, we compare our DCF with other state-of-the-art methods on these benchmarks. As expected, sampling more frames can further improve the accuracy, but also increase the FLOPs.

| Method | Backbone | Pretrain | Frames | GFLOPs | Sth-Sth V1 | |
|---|---|---|---|---|---|---|
| | | | | | Top1 | Top5 |
| TSN | BNInception | ImageNet | 8 | 16 | 19.5 | - |
| TRN-Multiscale | BNInception | ImageNet | 8 | 33 | 34.4 | - |
| R(2+1)D* | ResNet50 | ImageNet | 8 | 33 | 46.8 | 74.7 |
| TSM | ResNet50 | ImageNet | 8+16 | 98 | 49.7 | 78.5 |
| TANet | ResNet50 | ImageNet | 8+16 | 99 | 50.6 | 79.3 |
| SmallBigNet | ResNet50 | ImageNet | 8+16 | 157 | 50.4 | 80.5 |
| CorrNet | ResNet101 | - | 32 | 224x30 | 51.7 | - |
| Yang *et al.* | ResNet101 | - | 32 | 150x30 | 52.8 | - |
| TDN | ResNet101 | ImageNet | 16+64 | 132 | 55.3 | 83.3 |
| X3D* | X3D-S | - | 13 | 2.0 | 44.6 | 74.4 |
| X3D* | X3D-M | - | 16 | 4.7 | 47.3 | 76.6 |
| DCF (Ours) | X3D-S | - | 13 | 2.2 | 47.6 | 76.3 |
| DCF (Ours) | X3D-M | - | 16 | 5.1 | 49.5 | 78.5 |
| DCF (Ours) | ResNet50 | ImageNet | 8 | 35.9 | 50.5 | 79.4 |
| DCF (Ours) | ResNet50 | ImageNet | 16 | 71.9 | 54.1 | 82.0 |
| DCF (Ours) | ResNet101 | ImageNet | 16 | 131.3 | **55.8** | **84.3** |
| **Method** | **Backbone** | **Pretrain** | **Frames** | **GFLOPs** | **Sth-Sth V2** | |
| | | | | | Top1 | Top5 |
| TSM | ResNet50 | ImageNet | 16 | 86×6 | 63.4 | 88.8 |
| SmallBigNet | ResNet50 | ImageNet | 8+16 | 157 | 63.3 | 88.8 |
| TANet | ResNet50 | ImageNet | 16 | 86×6 | 64.6 | 89.5 |
| TAda2D | ResNet50 | ImageNet | 16 | 86×6 | 65.6 | 89.2 |
| TDN | ResNet101 | ImageNet | 16+64 | 132 | 66.9 | 90.9 |
| TimeSformer-HR | ViT-B | ImageNet | 16 | 1703×3 | 62.2 | - |
| MViT | ViT-B | Kinetics400 | 16 | 70.5×3 | 64.7 | 89.2 |
| X3D* | X3D-M | - | 16 | 4.7 | 59.3 | 86.1 |
| DCF (Ours) | X3D-M | - | 16 | 5.1 | 63.4 | 88.7 |
| DCF (Ours) | ResNet50 | ImageNet | 16 | 71.9 | 65.5 | 90.1 |
| DCF (Ours) | ResNet101 | ImageNet | 16 | 131.3 | **67.5** | **91.1** |

Table 4: Comparison with the state-of-the-art methods over action recognition on Something-Something V1 and V2 validation set. The results of R(2+1)D ResNet and X3D baselines are trained with the same training protocols for a fair comparison, which are marked with *.

| Method | Backbone | Pretrain | Frame | GFLOPs | Top1 | Top5 |
|---|---|---|---|---|---|---|
| TSN | InceptionV3 | ImageNet | 25 | 3.2×250 | 72.5 | 90.2 |
| TSM | ResNet50 | ImageNet | 16 | 86×30 | 74.7 | 91.4 |
| CorrNet | ResNet50 | - | 32 | 115×10 | 77.2 | - |
| TDN | ResNet50 | ImageNet | 16+64 | 94×30 | 77.5 | 93.2 |
| TAda2D | ResNet50 | ImageNet | 16 | 86×30 | 77.4 | 93.1 |
| X3D | X3D-M | - | 16 | 6.2×30 | 76.0 | 92.3 |
| SlowFast | ResNet50 | - | 8+32 | 65.7×30 | 77.0 | 92.6 |
| MViT | ViT-B | - | 16 | 70.5×5 | 78.4 | 93.5 |
| DCF (Ours) | X3D-M | - | 16 | 6.8×30 | 77.2 | 92.7 |
| DCF (Ours) | ResNet50 | ImageNet | 16 | 93.9×30 | 77.4 | 93.1 |

Table 5: Comparison with the state-of-the-art methods over action recognition on Kinetics-400 validation set. We instantiate our DCF with the backbones of R(2+1)D ResNet and X3D for evaluation.

For fair comparison with previous methods, we use 1 clip and center crop testing scheme on the Something-Something dataset and 10 clips and 3 crops for testing on the Kinetics- 400 dataset.

**Results on Something-Something.** We first validate the efficiency and effectiveness of the proposed DCF on Something-Something V1&V2. Table 4 shows the results and computation budgets (*i.e.*, number of GFLOPs) of other methods: temporal modeling based on 2D CNN methods ( TSM(Lin et al., 2019), TANet(Luo & Yuille, 2019), CorrNet(Wang et al., 2020), (Yang et al., 2020b), TDN(Wang et al., 2021), TAda2D(Huang et al., 2021)), 3D CNN architectures ( I3D(Carreira & Zisserman, 2017), SmallBigNet(Li et al., 2020a) and X3D(Feichtenhofer, 2020)) and Transformer

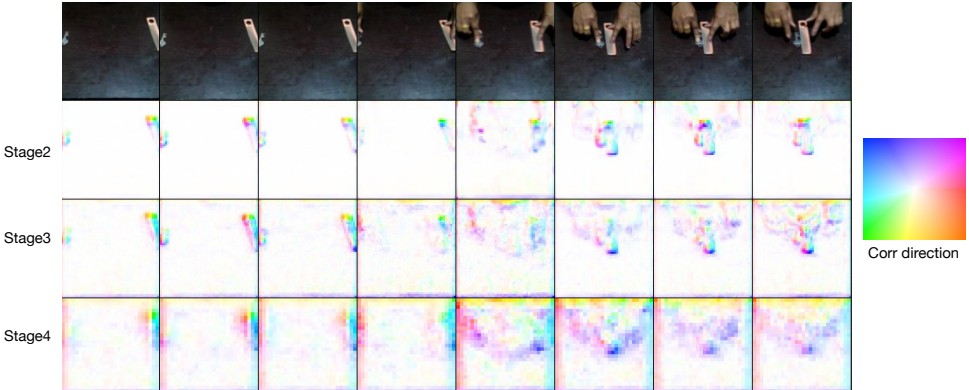

Figure 4: An example of the correlation pyramid on the SSV1 dataset. We provide a color wheel for correlation direction tendency on the right. The tendency of the correlation feature is consistent with the correct motion pattern. Note that the correlation from the early stage preserves fine details. Best viewed in color.

methods ( TimeSformer-HR(Bertasius et al., 2021) and MViT(Fan et al., 2021)). Under the same input frames and the same backbone of ResNet50, DCF consistently outperforms other methods with a comparable computation budget on both Something V1&V2. When compared to 2D CNN with temporal modules for all stages, our DCF consistently outperforms them on both datasets, demonstrating the effectiveness of DCF in motion modeling for action recognition. When compared to more recent 3D CNNs and Transformer methods, our DCF can still obtain slightly better performance than those methods. CorrNet (Wang et al., 2020) shares a similar with ours, namely the construction of correlation operator to establish frame-to- frame matches over convolutional feature maps to capture motion information. In particular, our DCF achieves better performance than CorreNet even with fewer input frames on SSV1. This result demonstrates that the proposed dense correlation fields make it better to model coarse-to-fine motion information.

**Results on Kinetics-400.** We also compare DCF to other state-of-the-art methods on Kinetics-400. DCF with R(2+1)D achieves 77.4% Top-1 accuracy, and it shows better performance than the state-of-the-art temporal modeling methods. DCF with X3D also surpasses 3D CNNs methods, including Slowfast(Feichtenhofer et al., 2019) and X3D(Feichtenhofer, 2020). Our best result is on par with the previous best performance on the Kinetics dataset.

### 4.4 Visualization of Correlation Feature

In this section, we visualize the correlation fields to better understand our method. Visualization results are shown in Figure 5. A color wheel on the right indicates the correlation direction tendency. Note that the correlation is incomplete at image boundary because of the image padding before the correlation operator. The correlation exhibits reliable motion patterns for the correct actions. For example, the key moves right (correlation color is red-purple), while the ruler moves left (correlation color is green-blue). The correlation pyramid shows that lower stage can preserve considerably more details and more localization information as it is downsampled fewer times. It motivates us to combine low-resolution, semantically strong correlations with high-resolution, semantically weak correlations.

## 5 Conclusion

This work proposes Dense Correlation Fields (DCF) for explicitly model motion information, which computes frame-to-frame similarity and builds up temporal-spatial correlation fields with coarse-to-fine strategy. DCF is unique in that it operates on a spatial correlation hierarchy that associates motion feature across resolutions and semantic levels. DCF is complementary to increasing the model capacity for motion modeling as a stand-alone and plug-in module. Our method outperforms the state-of-the-art temporal modeling 2D CNNs methods as well as 3D CNN methods in terms of motion modeling. We hope this work can facilitate further research in motion modeling.

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

APPENDIX

## A  MODEL STRUCTURE

| Stage | R(2+1)D-50 | | Output size |
|---|---|---|---|
| data | | | T×224×224 |
| stage1 | 1×7×7, 64, stride 1,2,2
pool, 1×3×3, stride 1,2,2 | | T×56×56 |
| stage2 | $\begin{bmatrix} 1 \times 1 \times 1, 64 \\ 3 \times 1 \times 1, 64 \\ 1 \times 3 \times 3, 64 \\ 1 \times 1 \times 1, 256 \end{bmatrix}$ | × 3 | T×56×56 |
| stage3 | $\begin{bmatrix} 1 \times 1 \times 1, 128 \\ 3 \times 1 \times 1, 128 \\ 1 \times 3 \times 3, 128 \\ 1 \times 1 \times 1, 512 \end{bmatrix}$ | × 4 | T×28×28 |
| stage4 | $\begin{bmatrix} 1 \times 1 \times 1, 256 \\ 3 \times 1 \times 1, 256 \\ 1 \times 3 \times 3, 256 \\ 1 \times 1 \times 1, 1024 \end{bmatrix}$ | × 6 | T×14×14 |
| stage5 | $\begin{bmatrix} 1 \times 1 \times 1, 512 \\ 3 \times 1 \times 1, 512 \\ 1 \times 3 \times 3, 512 \\ 1 \times 1 \times 1, 2048 \end{bmatrix}$ | × 3 | T×7×7 |
| global average pool, fc | | | # classes |

Table 6: The R(2+1)D ResNet-50 backbone for building DCF networks.

| Stage | X3D-S | | Output size |
|---|---|---|---|
| data | | | T×160×160 |
| stage1 | 1×3×3, 3×1×1, 24, stride 1,2,2 | | T×80×80 |
| stage2 | $\begin{bmatrix} 1 \times 1 \times 1, 54 \\ 3 \times 3 \times 3, 54 \\ 1 \times 1 \times 1, 24 \end{bmatrix}$ | × 3 | T×40×40 |
| stage3 | $\begin{bmatrix} 1 \times 1 \times 1, 108 \\ 3 \times 3 \times 3, 108 \\ 1 \times 1 \times 1, 48 \end{bmatrix}$ | × 5 | T×20×20 |
| stage4 | $\begin{bmatrix} 1 \times 1 \times 1, 216 \\ 3 \times 3 \times 3, 216 \\ 1 \times 1 \times 1, 96 \end{bmatrix}$ | × 11 | T×10×10 |
| stage5 | $\begin{bmatrix} 1 \times 1 \times 1, 432 \\ 3 \times 3 \times 3, 432 \\ 1 \times 1 \times 1, 192 \end{bmatrix}$ | × 7 | T×5×5 |
| global average pool, fc | | | # classes |

Table 7: The X3D-S backbone for building DCF networks.

Here we introduce the implementation of DCF for action recognition. We use the R(2+1)D(Tran et al., 2018) as 2D backbone network, and X3D(Feichtenhofer, 2020) as the 3D backbone network. Compared with the original R(2+1)D, we make a few changes to further simplify and improve its efficiency, *e.g.*, using depthwise temporal 1D-Conv, removing temporal downsampling. We provide the structure of R(2+1)D ResNet-50 in Table6 and X3D in Table7 for the reference. The dimensions of 3D output maps and filter kernels are in T×H×W, with the number of channels following. Residual blocks are shown in brackets. We use the output features of $stage2$, $stage3$, $stage4$ to build DCF, where they are spatially downsampled by respectively 4, 8 and 16 times, compared to the input frames. We omit $stage5$ as its spatial resolution is too low.

---

**Algorithm 1** The DCF block

---

**Parameters:** Pyramid output level $l$, temporal range $\tau$, patch size $p$.
**Input:** Video feature $F$, correlation volumes from the top-down pathway $C_{in}$.

1: $x = \text{Conv1}(F)$;
2: $C = \text{corr\_op}(x, \tau, p)$;
3: **if** $C_{in}$ is not None **then**
4:     corr\_pyramid = concat($C$, $C_{in}$);
5: **else**
6:     corr\_pyramid = $C$;
7: **end if**
8: $C' = \text{BN2}(\text{Conv2}(\text{corr\_pyramid}))$;
9: $F' = F + C'$;
10: $F' = \text{ReLU}(F')$;
11: $C_{out} = \text{List}()$;
12: **for** $i = 1$ to $l$ **do**
13:     $C = \text{Avg\_pool}(C, \text{kernel=2, stride=2})$; % in spatial dimensions;
14:     $C_{out}.\text{append}(C)$;
15: **end for**
16: Return $F', C_{out}$
**Output:** Video feature $F'$ enhanced by DCF, pyramid correlation volumes $C_{out}$.

---

## B  DCF Block Instantiation Details

We wrap the DCF into a plugin module that can be incorporated into many existing architectures. DCF is based on several correlation blocks. The design of correlation block is similar to Residual block (He et al., 2016). The current feature first undergoes a $1\times1\times1$ layer to reduce channel dimensions. And then we compute a correlation volume $C$ across frames. The correlation volume of current stage $C$ is combined with correlation volumes $C_{in}$ from the top-down pathway by concatenation. These correlation volumes are attached by a $1\times1\times1$ layer, a BN layer and a ReLU layer to produce the correlation feature $C'$. The video feature is merged by addition with the correlation feature. The current correlation volume is downsampled spatially and connected to the deeper stage for a pyramid correlation. The details is presented in Algorithm 1.

## C  Compitational Analysis

The correlation operator is intentionally designed to capture matching information between consecutive frames. The correlation operator uses dot-product as visual similarity and does not include learnable parameters. Consider the input feature with the shape of T × H × W × $C_{in}$, where $C_{in}$ denotes the number of input channels. The computation is:

$$\text{FLOPs(Corr)} = T \times H \times W \times C_{in} \times (K^2 \times 2L) \tag{4}$$

where K indicates the spatial region and L indicates the temporal range. In contrast to the 3D convolutions for spatiotemporal modelling, the computation is:

$$\text{FLOPs(3D Conv)} = T \times H \times W \times C_{in} \times C_{out} \times (K_t \times K_x \times K_y) \tag{5}$$

where $K_t, K_x, K_y$ indicate the filter sizes for temporal, height and width respectively. For the FLOPs, the computation of the correlation operator is at least an order of magnitude smaller than the standard 3D convolution term. A correlation layer constructs a 4D correlation volume with shape $T \times H \times W \times (K^2 \times 2L)$ for each level feature.

## D  Implementation Details

Here, we further describe the implementation details for action recognition. For a fair comparison, we keep all the training strategies the same for our baseline.

Our experiments on action recognition are conducted on three large-scale datasets. In the training scheme, we train with synchronized SGD using 16 GPUs for all action recognition models. Following common practice (Feichtenhofer et al., 2019; Wang et al., 2021), during training, each video frame is resized to have shorter side in $[256, 320]$ and a crop of $224 \times 224$ is randomly cropped. For all models, we use a dropout ratio (Hinton et al., 2012) of 0.5 before the classification heads. Temporally, we perform interval based sampling for Kinetics-400, with interval of 8 for 8 frames, interval of 6 for 13 frames and interval of 5 for 16 frames. On Something-Something dataset, we perform segment based sampling with segments of 8 or 16. We implement two kinds of testing scheme. When we consider the efficiency, 1-clip per video and the center crop of $224 \times 224$ is used for evaluation on SSV1 and SSV2 dataset. When pursuing high accuracy, we followed the common setting in (Wang et al., 2018b) to sample multiple clips per video and use three spatial crops of $256 \times 256$ on Kinetics-400 dataset. Note that X3D-S has a different resolution for training and testing.

In R(2+1)D backbone experiments, we use ResNet50 and ResNet101 initialized using ImageNet (Deng et al., 2009) pre-trained weights (He et al., 2016) to implement our DCF networks. We train the R(2+1)D model for 60 epochs on Something-Something dataset and 100 epochs on Kinetis-400 dataset. The batch size is 64 and the initial learning rate is 0.01 for Something-Something dataset. The batch size is 128 and the initial learning rate is 0.02 for Kinetis-400 dataset. We use half-period cosine schedule for decaying the learning rate. The weight decay is set to 5e-4 for Something-Something dataset and 1e-4 for Kinetis-400 dataset.

As for X3D backbone experiments, we use X3D-S and X3D-M to train our DCF networks from scratch. We train the X3D model for 128 epochs on Something-Something dataset and 256 epochs on Kinetis-400 dataset. The batch size is 128 and the initial learning rate is 0.4 for Something-Something dataset. The batch size is 256 and the initial learning rate is 0.4 for Kinetis-400 dataset. We use half-period cosine schedule for decaying the learning rate. The weight decay is set to 5e-5 for all datasets. For X3D-S models, we use a special resolution of $160 \times 160$ for training and testing.

# E  MORE VISUALIZATION EXAMPLES

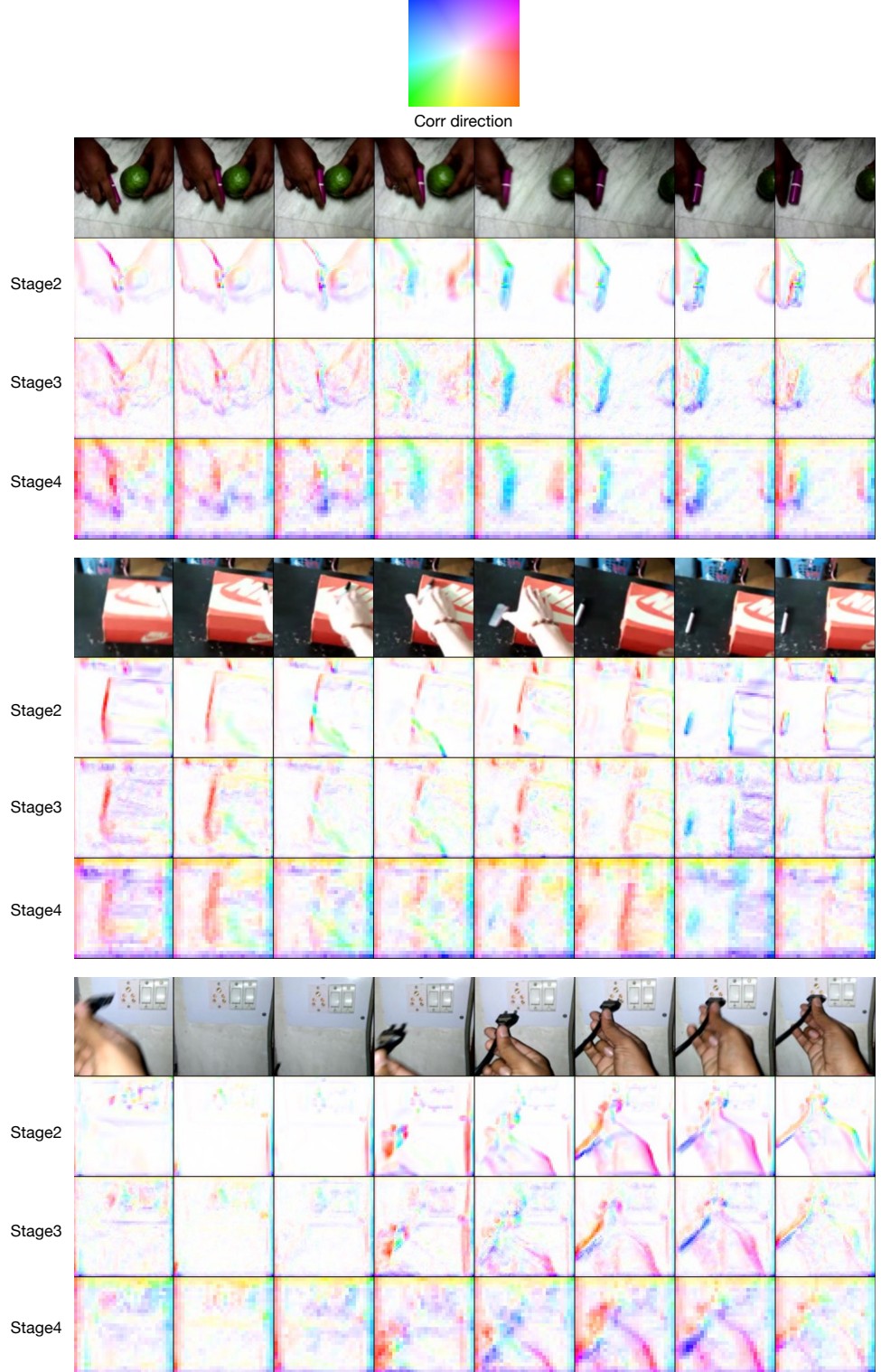

Figure 5: Examples of correlation pyramid on the SSV1 dataset. Best viewed in color.

