# OpenReview forum: "Dense Correlation Fields for Motion Modeling in Action Recognition"
_ICLR.cc/2023/Conference — Submitted to ICLR 2023_

### Official Review · Reviewer_bPjE · 2022-10-25

**Confidence:** 4
**Correctness:** 4
**Technical Novelty And Significance:** 3
**Empirical Novelty And Significance:** 4
**Recommendation:** 6

**Clarity, Quality, Novelty And Reproducibility:**

The implementation of the method is explained in detail and is clear to reader.
The paper is in good quality shape and also the idea seems novel. I am sure it is also easy to implement and reproduce this approach.

**Strength And Weaknesses:**

- DCF is a simple and novel idea that outperforms SOTA by focusing on motion in videos.
- Using spatial pyramid correlation feature
- DCF is stand-alone and can be applied to different backbones.

**Summary Of The Paper:**

This paper focuses on recognizing actions based on the motion information in the videos. The proposed idea (Dense Correlation Fields, DCF) which can be applied to different CNN backbones, computes frame to frame similarity to find spatial-temporal correlation fields.

**Summary Of The Review:**

The paper is pretty easy to follow and approaches the video action recognition from motion modeling point of view (which is not new, a lot of papers have leveraged differently from motion, either using Optical Flow or a different representation of the motion). I like this paper because it proposes a module that helps the network learn from motion at different scales (compared to SOTA which uses most likely a fixed representation of motion). It is a simple idea that works well.
My only concern about this method would be its computational time complexity and I would like to see a section on that to see how it performs comparing other approaches (specially since it captures motion at different stages in the network, my guess is it might make it computationally expensive).
Also, it would be nice to cite more related papers that took into account the motion modeling and representation, there are a lot, but here are some examples:
- V. Choutas, P. Weinzaepfel, J. Revaud, and C. Schmid. Potion: Pose motion representation for action recognition. In CVPR 2018, 2018.
- S. Sun, Z. Kuang, L. Sheng, W. Ouyang, and W. Zhang. Optical flow guided feature: a fast and robust motion represen- tation for video action recognition. In The IEEE Conference on Computer Vision and Pattern Recognition (CVPR), 2018.
- J. Jiang, Y. Cao, L. Song, S. Z. Y. Li, Z. Xu, Q. Wu, C. Gan,
C. Zhang, and G. Yu. Human centric spatio-temporal action
localization.
- L.Fan,W.Huang,S.E.ChuangGan,B.Gong,andJ.Huang.
End-to-end learning of motion representation for video understanding. In Proceedings of the IEEE Conference on Computer Vision and Pattern Recognition, pages 6016– 6025, 2018.
- Asghari-Esfeden, S., Sznaier, M. and Camps, O., 2020. Dynamic motion representation for human action recognition. In Proceedings of the IEEE/CVF Winter Conference on Applications of Computer Vision (pp. 557-566).
- C.-Y. Wu, M. Zaheer, H. Hu, R. Manmatha, A. J. Smola, andP.Kra ̈henbu ̈hl. Compressedvideoactionrecognition. In Proceedings of the IEEE Conference on Computer Vision and Pattern Recognition, pages 6026–6035, 2018

---

> ### Author Response · Authors · 2022-11-15
> **Response to Reviewer bPjE**
>
> Thanks very much for your constructive comments.
>
> **Q1: The computational cost.**
>
> **A1:**
> 1. The computation is also a focus in our work, as DCF is designed as a plug-in module with small extra computation overhead (less than 10% FLOPs compared to the backbone). Although we use DCF at different stages in the network, there is only a block in each stage. The computation of the correlation operator is an order of magnitude smaller than the standard 3D convolution term (detailed computation analysis and comparison with 3D Conv is presented in the Appendix). We also experiment with the different stage combinations to find an efficient form of DCF.
> 2. We also concern with the measured inference latency. We use an off-the-shelf CUDA implementation of correlation operation, which is widely used in the optical flow methods. We perform latency measurement on ResNet50 R(2+1)D baseline. Compared to the base model, our DCF brings 12% latency overhead with a 3.7% accuracy improvement on SSV1. We hope to provide a runtime analysis in another revision.
>
> **Q2: More related papers about motion modeling and representation.**
>
> **A2:**
> Thanks, we included these references.

---

### Official Review · Reviewer_NeMJ · 2022-10-26

**Confidence:** 4
**Correctness:** 4
**Technical Novelty And Significance:** 2
**Empirical Novelty And Significance:** 2
**Recommendation:** 3

**Clarity, Quality, Novelty And Reproducibility:**

The introduction of the proposed method can be further improved. I'd suggest to add one or two equations in Sec. 3.2 to make the description of long-term correlation and hierarchy aggregation more precise and clear.

As discussed in Weakness, the originality of the work is limited, and the paper doesn't sufficiently discuss how it is distinguished from the existing works on motion modeling.

**Strength And Weaknesses:**

Weakness
1. My major concern is the limited technical novelty and contribution of the paper. Most designs in DCF have considerable overlap with existing works: For example, the idea of using correlation at the feature level has been extensively studied in [1,2,3] and the idea of exploiting long-term motion and motion hierarchy has been discussed in [3]. The paper also doesn't discuss and compare with some important recent work on motion modeling, such as [2, 3, 4, 5].

2. The experiments in the paper is not convincing and the overall performance of DCF is not strong enough. First of all, many related works on motion model [2,3,4,5] are not compared in the paper, and many recent works on action recognition are not included. The visualization of learned motion patterns in Fig. 4 seem less appealing as those in [2, 3] as well.


[1] Wang, Heng, et al. "Video modeling with correlation networks." Proceedings of the IEEE/CVF Conference on Computer Vision and Pattern Recognition. 2020.
[2] Kwon, Heeseung, et al. "Motionsqueeze: Neural motion feature learning for video understanding." European conference on computer vision. Springer, Cham, 2020.
[3] Yang, Xitong, et al. "Hierarchical contrastive motion learning for video action recognition." BMVC 2021.
[4] Piergiovanni, A. J., and Michael S. Ryoo. "Representation flow for action recognition." Proceedings of the IEEE/CVF conference on computer vision and pattern recognition. 2019.
[5] Diba, Ali, et al. "Dynamonet: Dynamic action and motion network." Proceedings of the IEEE/CVF International Conference on Computer Vision. 2019.

**Summary Of The Paper:**

The paper proposes a new module, termed Dense Correlation Fields (DCF), to model motion information at feature level within neural networks. DCF involves a short-term module (i.e., correlation between adjacent frames), a long-term module (i.e., correlation between bidirectional and consecutive frames) and a spatial pyramid design (i.e. spatially downsampling correlation features for the next network stage). DCF can be applied to different backbone architectures and improves the baseline model on temporal modeling and action recognition. The paper conducts experiments on three common video action recognition datasets and provide ablation and visualization to analyze the impact of the DCF module.

**Summary Of The Review:**

The paper proposes a new module, DCF, for modeling motion information at the feature level. However, the technical novelty and contribution of the proposed method is not sufficient, and the experiments in the paper is not convincing enough. Therefore, I'd suggest "weak reject" to the paper.

---

> ### Author Response · Authors · 2022-11-15
> **Response to Reviewer NeMJ**
>
> Thank you for the review and suggestions.
>
> **Q1: Comparison with existing works.**
>
> **A1:**
> Regarding novelty: We agree that the overall framework draws inspiration from many existing works. However, the design of DCF is substantially novel. The main contribution of DCF is exactly that it pyramidally processes on a correlation hierarchy to build up a dense correlation field, which covers various motion patterns.
>
> CorrNet[1] and MSNet[2] compute frame-to-frame correlation over feature maps for effective motion estimation. Piergiovanni et al. [4] propose a flow layer that unrolls the iterations of the TV-L1 algorithm with learned parameters. DynamoNet[5] proposes dynamic motion filters by predicting the future frames to enrich motion representation. However, these methods learn motion representation at a single level, or multiple levels of the network individually, which may encounter a problem that local information disappears at spatially coarse levels.
>
> Yang et al. [3] also present a hierarchical method to bridge different levels in a network. This hierarchical design uses discriminative contrastive loss to enforce the motion features at high-level to predict the ones at low-level. However, this contrastive learning encourages features to have more similar representations, leading to less diversity to cover motion patterns. In our work, in contrast, we combine low-level features with high-level features via top-down and skip connections. In this way, we can preserve the diversity from the fine local information provided in the lower layer and the high-level semantic information from the deeper layer. Overall, DCF achieves better performance than Yang et al. 's method [3] on the SSV1 dataset (**DCF=54.1% top1 acc v.s. Yang et al.'s=52.8% top acc**).
>
> **Q2: The description of long-term correlation and hierarchy aggregation.**
>
> **A2:**
> Thank you for pointing this out. We've made some adjustments to the equation and tried to improve the description.
>
> ---
> We have made a more thorough comparison with these methods in the revised paper.  Please let us know if there are other works missing. We would be grateful if you could reconsider your score after taking into account the points above.

---

### Official Review · Reviewer_hBxH · 2022-10-28

**Confidence:** 1
**Clarity, Quality, Novelty And Reproducibility:** See above.
**Correctness:** 3
**Technical Novelty And Significance:** 2
**Empirical Novelty And Significance:** 3
**Recommendation:** 6

**Strength And Weaknesses:**

The following are more detailed comments and questions regarding the paper:
1, In Section 3.2, the paper suggests computing bi-directional correlation, L forward frames and L backward frames. This will lead to the correlation being computed multiple times for a given pair of frames. This may lead to a waste of computational cost. How does the paper handle this problem?

2, How to do padding for frames at the boundary of the sampled clip? Since now we are considering L frames, the padding may cause issues for the computation of correlation.

3, Figure 2 and Figure 3 (c) are not very illustrative.  I suggest explaining what exactly each operator is, and what are the sizes of input tensor and output tensor.

4, In section 4.1, how to use R(2+1)D with ResNet pretrained with ImageNet? Since R(2+1)D has separate spatial and temporal filters, how to initialize the filters with ImageNet pretrained ResNet?

5, In section 3.3, the paper suggests that L is set to L=2 and L=1 for the last stage. Will L=2 allow the model to long-term temporal information? It seems pretty short.


**Summary Of The Paper:**

The paper proposed dense correlation fields for action recognition. Based on CorrNet, the paper proposed to model temporal long-term correlation by computing correlation for frames that are far apart. The paper also designs a spatial hierarchical architecture to aggregate the information at different granularities. The paper presents results on the something-something V1 & V2, and Kinetics-400 dataset. The proposed method shows superior results than the original CorrNet.


**Summary Of The Review:**

See above.

---

> ### Author Response · Authors · 2022-11-15
> **Response to Reviewer hBxH**
>
> Thanks for the detailed feedback and we will further explain the question raised in the comment.
>
> **Q1: The problem of computing bi-directional correlation.**
>
> **A1:**
> Although it would lead to repeated correlation computation, the correlation volume contributes to each frame's motion representation. As with the attention operation in Transformer, correlation computes patch relations across frames. The relations in different correlation volumes focus on different types of motion information. We agree that the correlation computation repeats, but does not lead to a waste of computational cost.
>
> **Q2: How to do padding for frames at the boundary of the sampled clip?**
>
> **A2:**
> We use replicate padding to keep the video length. We experimented replicate padding and zero padding, while both of them work.
>
> **Q3: Illustration of Figure 2 and Figure 3 (c).**
>
> **A3:**
> Thank you for pointing this out. We tried to improve the illustration.
>
> **Q4: How to use R(2+1)D with ResNet pretrained with ImageNet?**
>
> **A4:**
> We initial the R(2+1)D with ResNet pretrained with ImageNet following TDN [1] strategy. More specifically, the spatial Conv inherits the pretrained ImageNet models, while the temporal Conv is initialized with Gaussian noise.
>
> **Q5: Long-term setting.**
>
> **A5:**
> As we use bi-directional correlation, the correlation is actually calculated across 4 frames when set L=2 (2 forward frames and 2 backward frames). As our intention was to enable motion modeling, we think that 4 frames are enough to cover a complete action. We construct an experiment with L=3 and it shows no further improvement.

---

### Official Review · Reviewer_SoQA · 2022-11-04

**Confidence:** 5
**Correctness:** 3
**Technical Novelty And Significance:** 2
**Empirical Novelty And Significance:** 3
**Recommendation:** 5

**Clarity, Quality, Novelty And Reproducibility:**

+ Clarity: The paper is well written.

+ Quality: good performance compared to baseline but the baselines are somewhat weak (see weakness 3).

+ Novelty: Not novel enough. See Weakness 1.

+ Reproducibility: According to the appendix, I think the results should be reproducible.

**Details Of Ethics Concerns:**

no.

**Strength And Weaknesses:**

### Pros
1. The idea is simple and the paper describes it in an organized way.

2. Nice visualization on correlation pyramid. It is an interesting finding that higher-level feature (low-resolution) has more semantically stong correlation.


### Cons
1. The idea of using cross-frame correlation is not new. The authors are suggested to discuss more papers which either construct a cost volume [a] or apply multiplicative interactions [b].

    [a] Zhao, et al. Recognize Actions by Disentangling Components of Dynamics. CVPR 2018

    [b] Wang, et al. Appearance-and-Relation Networks for Video Classification. CVPR 2018

2. The spatial correlation hierarchy is realized by "simply downsample"-ing. Have you tried other ways such as (1) conducting correlation operation on higher-level (lower spatial resolution) appearance feature map, or (2) a downsample version of the same-level feature map (i.e. downsample -> correlation instead of correlation -> downsample), or (3) conducting correlation on feature map with larger stride?

3. The baselines to compare with are weak and outdated.

    (1) Worse tradeoff than SlowFast. DCF w/ ResNet-50 is higher than SlowFast (R-50) by 0.4% but with 30% more GFLOPs. If the authors show an accuracy-GFLOPs tradeoff curve, I don't think DCF would have a comparable trade-off on par with with SlowFast compared to Slow-only.

    (2) There lacks a comparison to recent Transformer-based video models which remarkably boost the performance in the late few years. Also, I think the correlation operator is conceptually similar to the dot product operation in the self-attention. It makes me wonder if self-attention is a better way to capture such multiplicative relations and the gain of DCF compared to plain 2D/3D ConvNets might be simply diminishing.

**Summary Of The Paper:**

The paper proposes to use Dense Correlation Fields which is a pyramid of correlation volume of visual features. The correlation takes into account long-term correlation, and high-level semantically correlation feature. The method shows improvement over 2D CNN and 3D CNN baselines by a healthy margin.

**Summary Of The Review:**

Considering the limited novelty and the fact that improvements are based on a somewhat weak baseline, I don't think the paper meets the bar of ICLR.


===

See the latest comment for the final rating.

---

> ### Author Response · Authors · 2022-11-15
> **Response to Reviewer SoQA**
>
> Thanks for the detailed feedback and we will further explain the question raised in the comment.
>
> **Q1: Discussion with previous works.**
>
> **A1:**
> Thanks for bringing these two papers to our attention. Our work inherits the effectiveness of frame-to-frame matching in action recognition. [a] introduces cost volume processing and achieves good performance without the reliance on optical flow. [b] propose a learned relation branch by multiplicative interaction, which is integrated with an appearance branch. The main difference lies in the mechanism of integrating correlation. The observation from Figure.4 motivates us to combine the low-level correlation and high-level correlation. The principal advantage of DCF is that it produces a multi-scale motion feature representation in which all levels are spatially fine, including the low-resolution levels. In addition, DCF computes the correlation between consecutive frames, while they only apply the correlation between adjacent frames which will degrade performance. We will add this comparison to the paper.
>
> **Q2: The spatial correlation hierarchy.**
>
> **A2:**
> 1. **"correlation operation on higher-level (lower spatial resolution) appearance feature map".** We do conduct correlation operation on higher-level (lower spatial resolution) appearance feature map. As shown in Figure.2, we construct correlation volume for each spatial level. The correlation volumes in the low-level stages are down-sampled and connected to high-level stages. Please correct us if there is a misunderstanding.
> 2. **"A downsample version of the same-level feature map (i.e. downsample -> correlation instead of correlation -> downsample)".** It's a good question. The way of "downsample -> correlation" has a larger receptive field than the way of "correlation -> downsample". However, the price is less spatial detail. The motivation of DCF is to preserve fine local information from low-resolution level. So we chose the way of "correlation -> downsample". We've ever instantiated the way of "downsample -> correlation". The experiment shows that it is a 0.2% accuracy drop compared to the way of "correlation -> downsample". Note that "downsample -> correlation" costs more the computation.
> 3. **"Conducting correlation on feature map with larger stride".** We conducted an additional experiment with a larger stride. The performance is 0.3% worse (47.3% compared to 47.6%) than the original setting in the SSV1 dataset.
>
> **Q3: Comparison with other works.**
>
> **A3:**
>
> 1. **Comparison with SlowFast.** The sampling frame configuration is different between Slowfast and DCF on the K400 result in Table.5. Slowfast with R50 uses 8+32 frames while our DCF uses 16 frames. Empirically, sampling more frames can further improve accuracy. Since our initial motivation is motion modeling, we conducted additional experiments on the SSV2 dataset (the SSV2 size is on par with K400). The result is shown below. Compared with Slowfast with more frames, DCF achieves a better performance with fewer frames and computation. The improvement of DCF compared to baseline is also better than SlowFast compared to Slow-only.
>
> |        | Backbone  | Frames | GFLOPs | Top1@SSV2 |
> | ---- | ---- | ---- | ----| ----|
> | SlowOnly | R50 | 8 | 54.5 | 60.3 |
> | SLowFast | R50 | 8+32 | 65.7 (+20.6%) | 61.5 (+1.2) |
> | R(2+1)D | R50  | 8 | 32.7 | 60.2 |
> | DCF | R50 | 8 | 35.9 (+9.8%) | 62.9 (+2.7) |
>
> 2. **Comparison to recent Transformer-based video models.** We provide a comparison with Transformer methods in Table.4. We do agree that the correlation operator is conceptually similar to the dot product operation in the self-attention. However, self-attention computes relations between all the dimensions. It confuses the spatial dimension and temporal dimension. The correlation operator computes relations in temporal dimension and explicitly models motion information.
>
> ---
> We would be grateful if you could reconsider your score after taking into account the points above and addressing some of your concerns.

---

> > ### Comment · Reviewer_SoQA · 2022-12-05
> > **Response to authors**
> >
> > Thank the authors for providing the response and addressing my concerns. I think the ablations of hierarchy aggregation strengthen the paper. I would raise the overall score from 5 to 6 (marginally above the threshold) and empirical novelty score from 2 to 3. My only concern is the technical novelty is somewhat limited (agree with NeMJ in this point).

---

### Author Response · Authors · 2022-11-15
**Updated revision of our paper**

Dear commenters and reviewers,

thank you for your detailed critique of our paper. We have worked hard to revise our paper and address all of the points you have raised. Please check out the new version.
1. We improved the illustration of Figure 2 and Figure 3.
2. In Sec.2, we added more discussions with other relevant works mentioned by reviewers.
3. In Sec.3, we updated the description of long-term correlation and hierarchy aggregation.

Thank you - the authors.

---

### Comment · Reviewer_SoQA · 2022-12-10
**Post-reviewer-meeting response**

Dear AC and reviewers,

 Thank you AC for organizing the meeting. I'd like to reiterate my points as follows.

(1) The technical novelty is a bit weak and not sufficiently significant: Correlation-based operations have been proposed to explicitly model cross-frame motion for multiple computer vision tasks, e.g. optical flow [C1, C2] and video recognition [C3,C4]. The pyramid design for motion has also been found in [C5]. One may argue for the contribution of incorporating pyramid into motion modeling to "combine low-resolution, semantically strong correlation features with high-resolution, semantically weak correlation features to recover different motion patterns". The current version cannot fully justify this argument (eg. the evidence of Fig 4 is not strong enough).

(2) Empirical Novelty And Significance: I do appreciate that the authors provide fair comparison with SlowFast baseline and show the tradeoff is nontrivial. However, it should be noted that SlowFast is a paper in 2018 and the best backbone that is used in the paper is X3D which is published in 2020. Also X3D-M is used instead of the top-performing X3D-L/XL. Therefore, I see no evidence that the performance gain will hold for larger or modern backbones. Also it is worth noting that there have been multiple Transformer-based architectures in the last two years which attain very good performance on both K400 and sth-sth. As I stated in the comments, the self-attention operation is mathematically equivalent to correlation operation, potentially endowing the capability of modeling motion implicitly. Therefore, it would be better if the authors can provide further comparison with Transformer-based methods and try if such pyramid correlation design can further boost the Transformer-based baselines.

(3) Practical concerns. This is a point that came up to me during the reviewer meeting so I won't take this point into my final score (since it might be too late for the authors to respond). Computing high-resolution correlation feature is not a memory-efficient operation. I would imagine the memory consumption becomes a bigger issue if you try to build up a correlation pyramid. Given the same hardware constraint, will this memory overhead affect the number of batch size, thus reducing the effectiveness of parallelism, even if the GFLOPs do not significantly increase? Also, even if concatenation is a zero-FLOP operation, it does affect the runtime in real-world hardware. It might be better if we can take runtime speed into account besides theoretical GFLOPs. I think such analysis will bring empirical significance to the paper even if the technical significance might be mediocre.

To sum up, I don't see any critical flaw of this paper but the quality is slightly below the ICLR bar (i.e. 5). Any improvement on the technical novelty (i.e. more convincing evidence or justification of the motivation/observation) or the empirical novelty would strengthen this paper for next possible submission.

[C1] Sun, Deqing, et al. "Pwc-net: Cnns for optical flow using pyramid, warping, and cost volume." Proceedings of the IEEE conference on computer vision and pattern recognition. 2018.

[C2] Xu, Jia, René Ranftl, and Vladlen Koltun. "Accurate optical flow via direct cost volume processing." Proceedings of the IEEE Conference on Computer Vision and Pattern Recognition. 2017.

[C3] Zhao, Yue, Yuanjun Xiong, and Dahua Lin. "Recognize actions by disentangling components of dynamics." Proceedings of the IEEE Conference on Computer Vision and Pattern Recognition. 2018.

[C4] Wang, Heng, et al. "Video modeling with correlation networks." Proceedings of the IEEE/CVF Conference on Computer Vision and Pattern Recognition. 2020.

[C5] Yang, Xitong, et al. "Hierarchical contrastive motion learning for video action recognition." arXiv preprint arXiv:2007.10321 (2020).

---

### Comment · Reviewer_NeMJ · 2022-12-10
**Follow-up on the reviewers' meeting**

Thanks AC for organizing the meeting. Per our discussion, my concern about the insufficient technical contribution of the paper remains unchanged. I appreciate that the work extends the correlation-based operation to a pyramid design and manages to achieve better results. However, I do feel that the overall quality & impact of this paper is below the ICLR bar and readers cannot get new technical insights from this paper.

The idea of motion hierarchy and the effectiveness of spatial pyramid are well-aware to the research community. For example, [1,2,3] all discussed and studied leveraging motion information at multiple levels to improve action recognition. The submission proposes a different method to approach this idea (by re-using the downsampled motion features from the lower level), but does not address any significant technical challenges or provide new insights to this direction. Whether high-level motion are indeed well captured by the proposed method still cannot be justified, even given the visualization example in Figure4.

In all, I believe this submission needs improvement on the technical novelty and contribution to reach the bar of ICLR. I'll keep my original rating (3: reject).


[1] Ng, Joe Yue-Hei, et al. "Actionflownet: Learning motion representation for action recognition." 2018 IEEE Winter Conference on Applications of Computer Vision (WACV). IEEE, 2018.

[2] Wang, Heng, et al. "Video modeling with correlation networks." Proceedings of the IEEE/CVF Conference on Computer Vision and Pattern Recognition. 2020.

[3] Yang, Xitong, et al. "Hierarchical contrastive motion learning for video action recognition." BMVC 2021.

---

### Decision · Program_Chairs · 2023-01-20

**Decision:**

Reject

**Justification For Why Not Higher Score:**

After reading the paper and reviews and discussing with the three reviewers, the AC recommends the paper be rejected, due to the combination of minor technical differences combined with lack of compelling experimentation.  The authors are recommended to strength their work on these regards and resubmit.

**Justification For Why Not Lower Score:**

n/a

**Metareview: Summary, Strengths And Weaknesses:**

This paper proposes a plug-in module to correlate the flow features at different scales and resolutions for existing CNN backbones for action recognition.    Prior to discussion between the AC and reviewers, the reviews on this paper were mixed, ranging from 3, 6 and 8**.  The main concerns expressed by the reviewer (NeMJ and SoQA) were technical novelty and contribution; reviewer bPjE supports the paper and finds the ideas interesting albeit marginally different from existing work.

** Reviewer (hBxH) was not considered as the reviewer expressed low confidence and asked the AC to seek alternate opinions.

**Summary Of Ac-Reviewer Meeting:**


On the positive side, the proposed approach is simple and effective and achieves some performance gains.

On the down side, the submission leverages well-established concepts in video understanding, i.e. spatial pyramids, use of motion at various granularities (see existing works highlighted by SoQA and NeMJ).  While the reviewers appreciate the technical differences given in the author response, they feel that these differences are fairly minor. Similarly, the experiments, while sufficient to demonstrate effectiveness, do not provide new insight into the action recognition problem.

After discussion, the scores proposed by the reviewers (see final comments) were 3, 5 and 6.